# Fibroblast Growth Factor Receptor Inhibitors Decrease Proliferation of Melanoma Cell Lines and Their Activity Is Modulated by Vitamin D

**DOI:** 10.3390/ijms25052505

**Published:** 2024-02-21

**Authors:** Anna Piotrowska, Joanna I. Nowak, Justyna M. Wierzbicka, Paweł Domżalski, Monika Górska-Arcisz, Rafał Sądej, Delfina Popiel, Maciej Wieczorek, Michał A. Żmijewski

**Affiliations:** 1Faculty of Medicine, Department of Histology, Medical University of Gdańsk, Dębinki 1a, 80-384 Gdańsk, Poland; annapiotrowska@gumed.edu.pl (A.P.); j.chorzepa@gumed.edu.pl (J.I.N.); justyna.wierzbicka@gumed.edu.pl (J.M.W.); paweldomzalski@gumed.edu.pl (P.D.); 2Laboratory of Molecular Enzymology and Oncology, Intercollegiate Faculty of Biotechnology, University of Gdańsk and Medical University of Gdańsk, Dębinki 1, 80-384 Gdańsk, Poland; monika.gorska-arcisz@gumed.edu.pl (M.G.-A.); rafal.sadej@gumed.edu.pl (R.S.); 3Preclinical Development Departament, Celon Pharma S.A., Marymoncka 15, 05-152 Kazuń Nowy, Poland; delfina.popiel@celonpharma.com (D.P.); maciej.wieczorek@celonpharma.com (M.W.); 4Clinical Development Department, Celon Pharma S.A., Marymoncka 15, 05-152 Kazuń Nowy, Poland

**Keywords:** FGFR, FGFR inhibitors, melanoma, CPL304110, AZD4547, vitamin D, 1,25(OH)_2_D

## Abstract

Regardless of the unprecedented progress in malignant melanoma treatment strategies and clinical outcomes of patients during the last twelve years, this skin cancer remains the most lethal one. We have previously documented that vitamin D and its low-calcaemic analogues enhance the anticancer activity of drugs including a classic chemotherapeutic—dacarbazine—and an antiangiogenic VEGFRs inhibitor—cediranib. In this study, we explored the response of A375 and RPMI7951 melanoma lines to CPL304110 (CPL110), a novel selective inhibitor of fibroblast growth factor receptors (FGFRs), and compared its efficacy with that of AZD4547, the first-generation FGFRs selective inhibitor. We also tested whether 1,25(OH)_2_D_3_, the active form of vitamin D, modulates the response of the cells to these drugs. CPL304110 efficiently decreased the viability of melanoma cells in both A375 and RPMI7951 cell lines, with the IC50 value below 1 µM. However, the metastatic RPMI7951 melanoma cells were less sensitive to the tested drug than A375 cells, isolated from primary tumour site. Both tested FGFR inhibitors triggered G0/G1 cell cycle arrest in A375 melanoma cells and increased apoptotic/necrotic SubG1 fraction in RPMI7951 melanoma cells. 1,25(OH)_2_D_3_ modulated the efficacy of CPL304110, by decreasing the IC50 value by more than 4-fold in A375 cell line, but not in RPMI7951 cells. Further analysis revealed that both inhibitors impact vitamin D signalling to some extent, and this effect is cell line-specific. On the other hand, 1,25(OH)_2_D_3_, have an impact on the expression of FGFR receptors and phosphorylation (FGFR-Tyr653/654). Interestingly, 1,25(OH)_2_D_3_ and CPL304110 co-treatment resulted in activation of the ERK1/2 pathway in A375 cells. Our results strongly suggested possible crosstalk between vitamin D-activated pathways and activity of FGFR inhibitors, which should be considered in further clinical studies.

## 1. Introduction

Cutaneous melanoma, which is linked to 90% of skin cancer mortality, is considered the most dangerous form of skin tumour [1,2]. It should be underlined that the incidence of melanoma has constantly risen over the last five decades [3], although recent data offer some hope of stabilisation (3). At present, as forecasted by the American Cancer Society, melanoma is expected to be the fifth most common cancer in both males and females [4]. Since 2011, significant expansion of the path-breaking therapeutic arsenal, including targeted therapy with BRAF/MEK inhibitors and immunotherapy with immune checkpoint inhibitors cytotoxic T cell antigen 4 (CTLA-4) or programmed cell death protein 1 (PD-1), has been observed [5]. It resulted in a 5-year survival rate for metastatic melanoma of approximately 30% [4], which still is far from being satisfactory. However, before these new therapies were introduced, advanced patients could expect a 5-year survival rate of only 10% [6]. Strikingly, melanoma patients treated with targeted drugs indeclinably experience recurrence within a year of therapy initiation [7]. Up to 60% of patients treated with immune checkpoint inhibitors do not achieve any significant therapeutic response, and the development of acquired resistance in responders is often observed [8,9]. It should be emphasised that our understanding of the molecular nature of melanoma is still insufficient, which supports the necessity of developing novel adjuvant approaches to fight this fatal cancer.

The family of fibroblast growth factor receptors, which belong to receptor tyrosine kinases, comprises four transmembrane receptors, FGFR1-4, and one receptor, FGFR5, which lacks an intracellular domain [10,11]. More than twenty fibroblast growth factors (FGFs) are known as the ligands of FGFRs, of which eighteen may trigger dimerization of these receptors followed by an autophosphorylation of tyrosine residues in the cytoplasmic kinase domain [10,11]. Additionally, a repertoire of coreceptors, such as Klotho (KL) or cofactors, including heparin sulphate proteoglycans, influence interactions of FGFs with their receptors [10]. The kinase activity of FGFRs stimulates the downstream effector molecules, including phospholipase C gamma or fibroblast growth factor receptor substrate 2, leading to activation of multiple signalling pathways, among which protein kinase C (PKC), mitogen-activated protein kinase-extracellular signal-regulated kinase (MAPK-ERK), phosphoinositide 3-kinase/protein kinase B (PI3K/AKT), or STAT are involved in the regulation of cell proliferation, survival, migration, and invasion [10,11,12]. Aberrant activation of FGFRs is observed in 5–10% of all human cancers [11,13], with a 10–30% increase in urothelial carcinoma and intrahepatic cholangiocarcinoma [11]. Even though FGF/FGFR signalling is not considered as a leading factor driving melanoma development, as it is widely appreciated that it is a direct result of overactivation of the RAS/BRAF/MEK/ERK (MAPK) signalling pathway in this fatal skin neoplasm [10], it is suggested that activation of FGF/FGFR signalling in the complex repertoire of cells within the tumour microenvironment, including cancer-associated fibroblasts, in response to BRAF/MEK inhibitors, might be considered as one of the mechanisms that lead to the development of resistance to BRAF/MEK inhibitors, limiting the efficacy of targeted therapies [10]. It is also acknowledged that the survival and proliferation of melanocytes depend on FGF2, which is produced by neighbouring fibroblasts and keratinocytes in the skin [10,14,15]. The role of both FGF2 and FGFR1 signalling is suggested in melanoma progression [10,14]. Finally, it is postulated that the FGF/FGFR system plays a pivotal role in the modulation of the epithelial–mesenchymal transition in melanoma [16].

The expression of FGFR1 was documented in 67% of melanocytic nevi, with an observed increase up to 86% of primary melanomas [17]. Similarly, the level of FGFR3 was significantly higher in melanoma than in the surrounding healthy tissues [18]. It was also documented that FGFR3 promotes the growth, colony formation, migration, and invasion abilities of melanoma A375 cells [18]. Interestingly, it was reported that 10% of melanoma tumours and cell lines harbour mutations in the fibroblast growth factor receptor 2 (FGFR2) gene and, what is more, the mutation pattern reflects those induced by UV radiation. [14].

The aim of the present study was to characterise the cellular response of A375 and RPMI7951 melanoma lines to CPL304110, a novel selective inhibitor of fibroblast growth factor receptors (FGFRs) [19]. This drug is currently being tested in a clinical trial in adults with advanced solid malignancies (NCT04149691). An AZD4547, which belongs to the first generation of selective FGFR inhibitors [11] and is currently in phase II of clinical trials in patients with solid tumours, including melanoma and lymphomas, that have progressed following at least one line of standard treatment (NCT02465060), as well as in urothelial carcinoma patients with *FGFR2/3* gene alterations (NCT05086666), was also tested. Additionally, we have also explored whether 1,25(OH)_2_D_3_ modulates the effectiveness of selected drugs, since we have shown that vitamin D and its analogue, calcipotriol, enhance the activity of a classical chemotherapeutic, dacarbazine, in human malignant A375 melanoma cell line [20]. Recently we also showed that 1,25(OH)_2_D_3_ and other vitamin D derivatives significantly augment the efficacy of cediranib, an oral tyrosine kinase inhibitor of VEGFR1-3, PDGFR, and c-KIT, against A375 and SK-MEL-28 melanoma cell lines, out of four human cell lines that were tested, as well as against patient-derived melanoma cells [21,22]. What is more, there is a clear link between FGFRs signalling and vitamin D homeostasis, as loss of function in FGFRs leads to deregulation of *Cyp27b1* and *Cyp24a1* and the induction of hypervitaminosis D in a mouse model [23]. Although vitamin D is most widely known as a regulator of calcium and phosphate homeostasis in our body, it is appreciated that the role of this hormone is much more complex [24,25,26]. Through its involvement in inhibition of tumour cell proliferation, sensitisation to apoptosis and chemotherapy, and induction of cell differentiation or inhibition of epithelial-to-mesenchymal transition, it exerts pleiotropic anticancer activities [25,27].

## 2. Results

### 2.1. FGFR Inhibitors Efficiently Decrease Melanoma Cells’ Viability—The Efficacy of 304-110 Is Modulated by Vitamin D in A375 Melanoma Cells

Both tested FGFR inhibitors, CPL304110 and AZD4547, effectively decrease A375 melanoma cell viability from as low as 0.3 µM for CPL304110 (*p* < 0.0001) and 0.6 µM for AZD4547 (*p* < 0.01), as determined by an SRB proliferation assay (Figure 1A). CPL304110 decreased A375 melanoma cell viability maximally by about 78% at a 10 µM concentration during 72 h incubation, while the calculated relative IC50 value was 0.336 µM (Figure 1A). AZD4547 decreased A375 melanoma cell viability maximally by about 90% at 10 µM concentration, and the calculated relative IC50 value was 1.623 µM (Figure 1A).

Next, the effect of our FGFR inhibitors on the viability of RPMI7951 human melanoma cells was analysed, which, in contrast with A375 primary melanoma cell line derived from skin, is the metastatic cell line [28]. This model allows us to monitor the response of melanoma cells at different stages of disease progression. CPL304110 and AZD4547 decreased RPMI7951 melanoma cell viability from as low as 0.3 µM concentration for CPL304110 (*p* < 0.01) and 1.25 µM for AZD4547 (*p* < 0.001). CPL304110 decreased RPMI7951 melanoma cell viability maximally by about 63% at 10 µM concentration during 72 h incubation, while the calculated relative IC50 value was 0.926 µM (Figure 1B). AZD4547 decreased RPMI7951 cell viability maximally 66% at 10 µM concentration, and the calculated relative IC50 value was 2.606 µM (Figure 1B).

Since 1,25(OH)_2_D_3_ decreases the viability of A375 [29] and RPMI7951 (Appendix A), the effect of 1,25(OH)_2_D_3_ on the efficacy of the tested FGFR inhibitors against melanoma cells was tested. We found that 1,25(OH)_2_D_3_ slightly enhanced the cytotoxic effect of CPL304110 against A375 melanoma cells during 48 h incubation by decreasing the IC50 value by more than 4-fold (Figure 2A), but this was not the case for AZD4547 (Figure 2C). The IC50 value calculated either for CPL304110, or for AZD4547 was barely changed by 1,25(OH)_2_D_3_ in RPMI7951 melanoma cells (Figure 2B,D).

### 2.2. FGFR Inhibitors Trigger G0/G1 Cell Cycle Arrest in A375 Melanoma Cells and Increase SubG1 Fraction in RPMI7951 Melanoma Cells, and Their Activity Is Modulated by Vitamin D

The mechanism of proliferation inhibition in A375 and RPMI7951 melanoma cells was explored by analysing the distribution of appropriate melanoma cells in various phases of the cell cycle by flow cytometry. The values are summarised in Table 1. The CPL304110 inhibitor at 5 µM triggered G0/G1 cell cycle arrest in A375 melanoma cells during 24 h incubation (77.26% vs. 73.01% in control cells, *p* < 0.0001, Figure 3A), which was accompanied by a proportional decrease in the percentage of cells in S phase of the cell cycle. Interestingly, simultaneous treatment of A375 melanoma cells with CPL304110 and calcitriol triggered G2/M cell cycle arrest (20.41% vs. 16.43% in control cells, *p* < 0.0001, Figure 3A), indicating an induction of apoptotic/necrotic cell death. Similar results were observed for AZD4547 at 5 µM concentration, which triggered G0/G1 cell cycle arrest during 24 h incubation (78.56% vs. 73.01% in control cells, *p* < 0.0001), although no modulation of this effect by vitamin D in our experimental conditions was observed (Figure 3B). During extended, 48 h incubation, CPL304110 induced A375 cell death, as observed by the increase in the SubG1 cell fraction (8.03% vs. 2.30% in control cells, *p* < 0.001), considered as apoptotic/necrotic cells (Figure 3C). However, simultaneous treatment of A375 cells with CPL304110 and vitamin D resulted in G2/M cell cycle arrest compared to treatment with CPL304110 alone (17.10% vs. 12.37% in cells treated with CPL304110 alone, *p* < 0.01, Figure 3C). Prolonged incubation of A375 melanoma cells with AZD4547 triggered G0/G1 cell cycle arrest (74.81% vs. 70.23% in control cells, *p* < 0.001, Figure 3D), similarly to 24 h incubation, but an increase in the SubG1 cell fraction was also observed (7.07% vs. 2.30% in control cells, *p* < 0.0001, Figure 3D).

In RPMI7951 melanoma cells, CPL304110 at 5 µM concentration increased the percentage of the SubG1 fraction (8.0% vs. 1.06% in control cells, *p* < 0.001, Figure 4A), while co-stimulation with 1,25(OH)_2_D_3_ resulted in G0/G1 cell cycle arrest after 24 h incubation (59.27% vs. 54.28% in control cells, *p* < 0.01, Figure 4A). AZD4547 used alone at 5 µM triggered the G0/G1 cell cycle arrest in RPMI7951 (61.18% vs. 54.28% in control cells, *p* < 0.0001, Figure 4B) and a slight increase in the percentage of the SubG1 cell fraction (3.53% vs. 1.06% in control cells, *p* < 0.05, Figure 4B), without any modulation by vitamin D. During 48 h incubation of RPMI7951 cells with CPL304110, a pronounced increase in the percentage of the SubG1 cell fraction was observed (19.81% vs. 1.94% in control cells, *p* < 0.0001, Figure 4C), whereas vitamin D increased the percentage of the cells in G0/G1 phase in comparison to this FGFR inhibitor alone (49.30% vs. 42.07%, *p* < 0.001, Figure 4C). An incubation of RPMI7951 cells with AZD4547 at 5 µM concentration for 48 h resulted in a similar pattern of cell distribution throughout the cell cycle phases as was observed for 24 h incubation; however, the increase in the percentage of SubG1 cell fraction was more pronounced (Figure 4D).

### 2.3. FGFR Inhibitors Modulate the Expression of FGFs, FGF23, α-KLOTHO and Vitamin D-Related Genes at mRNA Level

Appreciating the importance of *FGF/FGFR* signalling in oncogenic pathways [30], the expression of *FGFR1* and *FGFR2*, *FGF23* and *α-Klotho*, with the latter two involved in the endocrine signalling through *FGFRs* [31], was evaluated under experimental conditions used for in vitro tests. Additionally, since we observed a modulatory effect of 1,25(OH)_2_D_3_ on the activity of CPL304110 and AZD4547 as described above, we tested whether FGFR inhibitors will modulate the expression of vitamin D receptor (VDR) and selected vitamin D-associated genes. Although the CPL304110 did not influence the mRNA level for VDR in A375 melanoma cells (Figure 5A), it showed a decrease after simultaneous treatment with CPL304110 and 1,25(OH)_2_D_3_ (*p* < 0.01 vs. control and *p* < 0.05 vs. treatment with 304–110 alone, Figure 5A). AZD4547 treatment also decreased the mRNA level for VDR (*p* < 0.01); however, the effect was not further modulated by 1,25(OH)_2_D_3_ (Figure 5A). Neither CPL304110 nor AZD4547 alone had an impact on the expression of CYP27B1, 1α-hydroxylase of 25(OH)D_3_—a cytochrome essential for a final activation of 1,25(OH)_2_D_3_ [25]—however, we noticed a decrease in the mRNA level of CYP27B1 in A375 melanoma cells simultaneously treated with CPL304110 and vitamin D (Figure 5B). Consistent with our previous results [21,32], we observed a profound increase in the mRNA level for CYP24A1 in A375 melanoma cells treated with 1,25(OH)_2_D_3_ (Figure 5C; *p* < 0.001). A slight (greater than 2-fold) increase in the mRNA level for CYP24A1 was also observed in A375 melanoma cells treated with CPL304110 (Figure 5C, *p* < 0.05) and the effect was strongly enhanced by 1,25(OH)_2_D_3_. Interestingly, AZD4547 treatment resulted in a slight decrease in mRNA level for CYP24A1 (*p* < 0.05), while the addition of vitamins reversed that effect (Figure 5C). We did not observe any significant influence of our tested FGFR inhibitors on mRNA level for FGF23 (FGFR1 natural ligand [31]) in A375 cells under the experimental conditions used (Figure 5D), although we noticed a minor increase in the mRNA level in cells treated solely with 1,25(OH)_2_D_3_ (Figure 5D; *p* < 0.05). Both tested FGFR inhibitors decreased the mRNA level for FGFR1 (Figure 5E; *p* < 0.05 for 304–110 and *p* < 0.01 for AZD4547), with the effect being slightly enhanced by 1,25(OH)_2_D_3_ for AZD4547 (Figure 5E, *p* < 0.05 vs. treatment with AZD4547 alone). Interestingly, treatment of melanoma cells with CPL304110 or AZD4547 compounds did not affect the mRNA level of FGFR2 (Figure 5F). On the other hand, CPL304110 decreased the mRNA level for α-Klotho (KL) (Figure 5G, *p* < 0.001), an essential component of fibroblast growth factor (FGF) receptor complexes, which is required for FGF23 to bind to FGFR with high affinity [33]. The presence of mRNA for KL in A375 cells treated with AZD4547 was below the level of detectability (Figure 5G). Curiously, we noticed an increase in the mRNA level for KL in melanoma cells treated with 1,25(OH)_2_D_3_ alone (Figure 5G; *p* < 0.01) or in combination with tested FGFR inhibitors in comparison to inhibitors alone (Figure 5G).

In RPMI7951 melanoma cells treated simultaneously with each of our tested FGFR inhibitors and 1,25(OH)_2_D_3_, but not for a monotreatment, we observed an increase in the mRNA level for VDR (Figure 6A, *p* < 0.01 for CPL304110 and *p* < 0.01 for AZD4547). No significant effect of either CPL304110 or AZD4547 compounds was observed for the mRNA level of CYP27B1; however, we noticed a decrease in the mRNA level in RPMI7951 cells simultaneously treated with AZD4547 and 1,25(OH)_2_D_3_ (Figure 6B). As for CYP24A1 mRNA, as expected, we noticed a marked increase in RPMI7951 cells treated with 1,25(OH)_2_D_3_ (Figure 6C; *p* < 0.01). CPL304110 also induced an increase in the mRNA level for CYP24A1, with the effect being enhanced by an addition of 1,25(OH)_2_D_3_ (Figure 6C; *p* < 0.01 vs. monotreatment). The compound CPL304110 slightly decreased the mRNA level for FGF23 (Figure 6D; *p* < 0.05), with no additional influence of 1,25(OH)_2_D_3_. No significant influence on the mRNA level of FGFR1 in RPMI7951 melanoma cells was observed under the listed experimental conditions (Figure 6E). Most importantly, we noticed an increase in the mRNA level for FGFR2 in RPMI7951 melanoma cells treated with each of our tested FGFR inhibitors and 1,25(OH)_2_D_3_ (Figure 6F, *p* < 0.05 for both) or with 1,25(OH)_2_D_3_ alone (Figure 6F, *p* < 0.05), but not for monotreatment with CPL304110 nor AZD4547. Finally, we observed an increase in the mRNA level for KL in RPMI7951 cells treated with AZD4547 (Figure 6F, *p* < 0.05), with the effect being reversed by 1,25(OH)_2_D_3_.

### 2.4. 1,25(OH)_2_D_3_ Regulates Levels of FGFR1 and 2 Receptors, as Well as Activation of FGFRs and ERK1/2 in A375 Melanoma Cells Treated with FGFR Inhibitors

The protein levels of all four members of the fibroblast growth factor receptors superfamily, FGFR1-4, and their activation by phosphorylation at Tyr653/654, as well as downstream activation of ERK1/2 by phosphorylation at ERK1/2-Thr202/Tyr202, were investigated in A375 melanoma cells incubated for 24–72 h, with CPL304110 in the presence or absence of 1,25(OH)_2_D_3_. The protein level of FGFR1 declined over time in A375 cells treated with 1,25(OH)_2_D_3_ alone or with 1,25(OH)_2_D_3_ combined with CPL304110 (Figure 7). Interestingly, the decrease in the protein level of FGFR1 was profoundly exacerbated by an addition of 1,25(OH)_2_D_3_ (Figure 7). A tendency similar to the effect of 1,25(OH)_2_D_3_ was noticed for the protein level of FGFR2, although the decrease was not that substantial. Interestingly, an increase in the FGFR2 protein level was observed in melanoma cells treated with CPL304110, while an addition of 1,25(OH)_2_D_3_ not only reversed this effect, but vastly decreased the protein level of FGFR2 compared to the untreated cells (Figure 7). The protein products for FGFR3 and 4 were not detected under the experimental conditions used. The activation of FGFRs by assessing the FGFRs phosphorylation at Tyr653/654, which are tyrosines found in the activation loop of these receptors [34], was also assessed. Interestingly, in A375 melanoma cells, CPL304110 treatment increased the FGFR2 protein level and its phosphorylation, while addition of 1,25(OH)_2_D_3_ very strongly decreased the level of both, and the effect was time dependent (Figure 7). The protein level of ERK1/2, one of the downstream effectors of FGFRs [10], increased with time in A375 melanoma cells treated with 1,25(OH)_2_D_3_. The ERK1/2 protein level also increased in cells treated with CPL304110, and the effect was time independent. An additional slight increase in the protein level after 24 h was noticed in cells simultaneously treated with CPL304110 and 1,25(OH)_2_D_3_ (Figure 7). Since the phosphorylation of ERK1/2 at tyrosine and threonine is required for enzyme activation [35], the level of Thr202/Tyr202-ERK1/2 was also assessed. The protein level of Thr202/Tyr202-ERK1/2, was increased by either 1,25(OH)_2_D_3_ or CPL304110 (Figure 7), while the increase was most pronounced in cells simultaneously treated with CPL304110 and 1,25(OH)_2_D_3_ for 72 h. Finally, the impact of the experimental treatment of 389 A375 cells on the protein level of vitamin D receptor (VDR), along with its downstream effector CYP24A1, was also assessed. As expected, the levels of both VDR and CYP24A1 proteins increased in cells treated with 1,25(OH)_2_D_3_ (Figure 7). The effect was time dependent, and was most pronounced after 48 h. An increase in the VDR protein level was also observed in A375 melanoma cells treated simultaneously with 1,25(OH)_2_D_3_ and CPL304110. Interestingly, although an increase in the protein level of CYP24A1 was observed for both CPL304110 and combined use of CPL304110 and 1,25(OH)_2_D_3_, for the latter, a slight decrease was observed compared to that observed for treatment with CPL304110 alone (Figure 7).

## 3. Discussion

Considering the high expression and the rate of mutations of FGFRs in melanomas, we proposed an anti-FGFR therapy that is highly relevant to explore in this aggressive skin neoplasm. The effect of the novel FGFR inhibitor, CPL304110, that is currently in phase I of clinical trials in adults with advanced solid malignancies (NCT04149691), and AZD4547, the first-generation selective FGFRs inhibitor, was tested in primary and metastatic melanoma cell lines, A375 and RPMI7951, respectively [28]. That approach allowed us to monitor the response of melanoma cells to the tested compounds throughout the disease’s progression.

We found both FGFR inhibitors, CPL304110 and AZD4547, highly efficient against proliferation of A375 melanoma line, especially during prolonged 72 h incubation, as we observed about 78% and 90% inhibition of cell proliferation by these compounds, respectively. The calculated IC50 value below 1 µM (IC50 = 0.336 µM) shows that the A375 melanoma cell line is more sensitive to our novel FGFRs inhibitor, CPL304110, than to AZD4547 (IC50 = 1.623 µM). Interestingly, vitamin D slightly enhanced the cytotoxic effect of CPL304110 at its lower concentrations during shorter, 48 h incubation, by decreasing the IC50 value by more than 4-fold for this FGFR inhibitor (IC50 = 0.966 µM versus 0.210 µM with 1,25(OH)_2_D_3_). A similar tendency, an enhancement of the antiproliferative effects of AZD4547 by 1,25(OH)_2_D_3_, exemplified by lowering of the IC50 value, was recently observed in human luminal B breast cancer cell line BT-474 [36].

In contrast to previously documented experiments related to selected non-small cell lung carcinoma lines, in which highly invasive NCI-H1703 cells strongly responded to CPL304110 [12], our malignant RPMI7951 melanoma cell line was apparently less sensitive to the tested compounds than the A375 line, since the calculated IC50 remained above or close to 1 µM during 48 h incubation. What is more, prolonged 72 h incubation did not result in further decrease in the IC50 value for CPL304110. Very interestingly, we also noticed that our two melanoma cell lines responded differently to the active form of vitamin D, 1,25(OH)_2_D_3_ in terms of the change in the mRNA levels of *FGFR2* in the experimental conditions used. Treatment of metastatic RPMI7951 melanoma cells with 1,25(OH)_2_D_3_ triggered an increase in the mRNA level for *FGFR2*, while it had no effect in A375 primary melanoma cells. Apparently, the observed differences reflect the distinct genetic background of the melanoma cell lines used in the study, which, in turn, potentially reflects the disease progression [28]. On the other hand, resistance to FGFR inhibitors may develop as a result of receptor mutation or hyperactivation of alternative signalling that surpasses their inhibitory effects [11].

We have shown that the tested FGFR inhibitors, CPL304110 and AZD4547, trigger G0/G1 cell cycle arrest in A375 melanoma cells and increase the SubG1 fraction, composed of apoptotic/necrotic cells, in RPMI7951 melanoma cells. Induction of G0/G1 cell cycle arrest by CPL304110 was previously observed in NCI-H1581 and NCI-H1703 lung cancer lines [12]. Additionally, our observations of melanoma cell death induction after treatment with CPL304110 or AZD4547 are in an agreement with other published data, which described the induction of apoptosis after inhibition of the FGFR system [37,38]. Interestingly, 1,25(OH)_2_D_3_ modulated the activity of CPL304110, but not AZD4547, in relation to the regulation of the cell cycle in melanoma cells, since we observed G2/M rather than G0/G1 cell cycle arrest in A375 cells and G0/G1 cell cycle arrest rather than an induction of SubG1 fraction in RPMI7951 melanoma cells compared to the monotreatment.

It seems that 1,25(OH)_2_D_3_ may affect the activity of the novel FGFRs inhibitor, CPL304110, by decreasing the FGFR1 and, even more profoundly, the FGFR2 protein level and activation, since we determined a significant decrease in the protein level of FGFRs phosphorylated at Tyr653/654. This process is essential to the activity of tyrosines found in the activation loop of these receptors [34]. Surprisingly, even though we expected a downward trend, CPL304110 treatment triggered an increase in the ERK1/2 protein level, as well as its phosphorylation. However, curiously, in lung cancer CPL304110-resistant cell lines, no changes in expression level or phosphorylation were observed for ERK compared to corresponding sensitive variants [12]. Also, we observed an effect of 1,25(OH)_2_D_3_ on one of their downstream effectors, ERK1/2, and a decrease in the level of Thr202/Tyr202-ERK1/2. The upregulation of FGFR2 expression at protein level is associated with poor prognosis in human breast cancer [39]. FGFR2 signalling was found to promote human gastric cancer progression through the downregulation of TSP4 via the PI3K-AKT-mTOR pathway [40]. Additionally, negative correlation between FGFR2 expression and patient outcome was found for diffuse-type gastric cancer [41]. High levels of FGFR-2 IIIC isoform, mostly expressed in mesenchymal cells, contributed to disease aggressiveness in pancreatic ductal adenocarcinoma and imparted pancreatic tumour cells with cancer stem cell features [42]. On the other hand, Ohashi’s study suggested that the decrease or loss of FGFR2 in high-grade gliomas correlates with a high proliferation rate of the tumour and a poor prognosis [43]. Apparently, FGFR2 cancer signalling might be context dependent. Importantly, we also noticed the upregulation of the VDR and a slight downregulation of CYP24A1 protein level in melanoma cells simultaneously treated with 1,25(OH)_2_D_3_ and CPL304110 compared to monotreatment with the novel FGFRs inhibitor. It seems that in the experimental conditions used, catabolism of vitamin D by CYP24A1, the 24-hydroxylase, which is downstream effector of VDR responsible for 1,25(OH)_2_D_3_ final deactivation [27], is decreased.

In conclusion, we documented for the first time (to our knowledge) that CPL304110, a novel selective FGFR inhibitor, is highly efficient against the proliferation of primary A375 and metastatic RPMI7951 melanoma cell lines, especially during prolonged incubation, although malignant RPMI7951 melanoma cell line is less sensitive to the tested compounds. 1,25(OH)_2_D_3_ slightly, but significantly, enhanced the cytotoxic effect of CPL304110, decreasing the IC50 value by more than 4-fold for this FGFR inhibitor in the A375 cell line. The effect was far less pronounced in RPMI7951 melanoma cells. Most notably, 1,25(OH)_2_D_3_ sensitised A375 melanoma cells to the novel FGFRs inhibitor, CPL304110, at the posttranscriptional level, decreasing the FGFR1 and 2 protein level as well as their activation.

## 4. Materials and Methods

### 4.1. Chemicals

1,25(OH)_2_D_3_ was purchased from Sigma Aldrich (Merck KGaA, Darmstadt, Germany). CPL304110 was provided by Celon Pharma S.A. (Kazuń Nowy, Poland) [19]. AZD4547 was purchased from AstraZeneca.

### 4.2. Cell Culture

Human A375 cell line (CRL-1619) and RPMI7951 (HTB-66) were purchased from the American Type Culture Collection (Manassas, VA, USA). A375 cells were cultured in Dulbecco’s modified Eagle’s medium (DMEM, Sigma Aldrich; Merck KGaA, Darmstadt, Germany) supplemented with 10% foetal bovine serum (FBS) (Biological Industries, Beit Haemek, Israel) and 1% penicillin/streptomycin (Sigma Aldrich; Merck KGaA, Darmstadt, Germany) in an incubator with 5% CO_2_ at 37 °C. RPMI7951 cells were cultured in Minimum Essential Medium Eagle, with Earle’s salts and non-essential amino acids (MEM, Sigma Aldrich; Merck KGaA), supplemented with 10% foetal bovine serum (FBS) (Biological Industries, Beit Haemek, Israel), 1% penicillin/streptomycin (Sigma Aldrich; Merck KGaA, Darmstadt, Germany), 1 mM sodium pyruvate, and 2 mM L-glutamine (Sigma Aldrich; Merck KGaA, Darmstadt, Germany both). Appropriate medium supplemented with 2% charcoal stripped FBS was used for all procedures, and the effects of 1,25(OH)_2_D_3_ were examined.

### 4.3. Viability Assay

The sulforhodamine B (SRB) assay was performed according to the previously described procedure [21,32]. Briefly, the human melanoma A375 or RPMI7951 cells were seeded in 96-well plates (3000 cells per well), cultured overnight and treated simultaneously with serial dilutions of CPL304110 or AZD4547 (0.15–10 µM) and 1,25(OH)_2_D_3_ at 100 nM concentration, and incubated for an additional 48–72 h. Following fixation with 10% trichloroacetic acid at 4 °C for 1 h, the cells were washed (5× with distilled water) and the staining solution comprising 0.4% SRB (Sigma Aldrich; Merck KGaA, Darmstadt, Germany) in acetic acid was added to each well for 15 min. Finally, the plates were washed with 1% acetic acid and air-dried at RT overnight. The SRB dye was solubilised with a solution of 10 mM buffered Tris Base (pH 10.5) and the absorbance was measured at 570 nm using an Epoch™ microplate spectrophotometer (BioTek Instruments, Inc., Winooski, VT, USA).

### 4.4. Cell Cycle Analysis

The cell cycle status was analysed with flow cytometry, based on quantification of DNA content, as described previously [21]. Next, A375 and RPMI7951 melanoma cells were treated with 1,25(OH)_2_D_3_ at 100 nM concentration, CPL304110 or AZD4547 at 5 µM concentration or with a combination of vitamin D with the selected FGFR inhibitor for 24 or 48 h. Trypsinized human malignant melanoma cells, as well as cells from the culture medium, were fixed together in 70% ethanol for 24–48 h at 4 °C. Following treatment with ribonuclease, the DNA was stained with propidium iodide (PI; Sigma Aldrich; Merck KGaA) for 30 min at 37 °C. The fluorescence of the PI-stained cells was measured by flow cytometry (FACSCalibur™; Becton, Dickinson and Company, Franklin, Lakes, NJ, USA). The results were analysed using the CellQuest™ Pro Software version 6.0 (Becton, Dickinson and Company) and expressed as a percentage of cells with DNA content corresponding to apoptotic/necrotic cells (subG1 fraction) or cells in G1, S and G2/M phases of the cycle.

### 4.5. RT-PCR

A375 and RPMI7951 human melanoma cells were treated with 1,25(OH)_2_D_3_ at 100 nM concentration, CPL304110 or AZD4547 at 5 µM concentration, or with a combination of vitamin D with the selected FGFR inhibitor for 24 h. Using the ExtractME^®^Total RNA Kit (Blirt, Poland, EM09.1-250) total RNA was extracted according to the manufacturer’s instructions. The concentration and purity of isolated RNA was measured by an EpochMicroplate Spectrophotometer (BioTek, Winooski, VT, USA). Extracted RNA was reverse transcribed and cDNA synthesised using a RevertAid™ First Strand cDNA Synthesis Kit (Thermo Fisher Scientific Inc., Waltham, USA). Real TimePCR was performed using a StepOnePlus™ Real-Time PCR System (LifeTechnologies-Applied Biosystems, Grand Island, NY, USA) with RealTime AMPLIFYME SYBR™ Green No-ROX Mix (Blirt, Poland, AM01). All primers were purchased from Sigma-Aldrich (Merck KGaA, Darmstadt, Germany). The expression was normalised by the comparative ΔΔ-Ct method, using RPL37A as a housekeeping gene, followed by calibration (fold change calculation) to normalised expression data of samples from control (ratio = 1). Dynamic melting curve analysis was performed for all reactions to ensure specificity of the PCR amplification. Primer sequences are summarised in Table 2.

### 4.6. Immunoblotting

A375 melanoma cells were treated with 1,25(OH)_2_D_3_ at 100 nM concentration, CPL304110 at 5 µM concentration, or with a combination of vitamin D with CPL304110 for 24 h. The cells were scraped and subsequently lysed in the presence of ice-cold RIPA buffer (Sigma Aldrich; Merck KGaA) supplemented with protease inhibitor cocktail. Protein concentrations were determined by the Bradford assay. The equal amounts of protein (~20 μg) were separated by SDS-PAGE, followed by transfer onto a nitrocellulose membrane. The membranes were blocked for 1 h in 5% skimmed milk in TBS-T and probed with specific primary antibodies at 4 °C. The antibodies against FGFR1 (#9740T), FGFR2 (#23328), FGFR4 (#D3B12), FGFR-Tyr653/654 (#3471), ERK1/2 (#9102), and ERK1/2-Thr202/Tyr203 (#9101) were obtained from Cell Signalling Technology (Danvers, MA, USA). The antibodies resistant to FGFR3 (sc-13121), VDR (sc-13133) and CYP24A1 (sc-66851) were from Santa Cruz Biotechnology. Appropriate secondary Alexa Fluor^®^-conjugated antibodies (680 or 790 nm) (Jackson ImmunoResearch, #111-625-144, #715-655-150) and Odyssey^®^ CLx imaging system (LI-COR^®^ Biosciences, NE, USA) were used to detect protein bands for FGFR-2 and FGFR-Tyr653/654. For FGFR1,3 and 4, ERK1/2, ERK1/2-Thr202/Tyr203, VDR and CYP24A1 secondary goat anti-rabbit or bovine anti-mouse antibodies conjugated to peroxidase (Sigma Aldrich; Merck KGaA #A0545 or Santa Cruz Biotechnology sc-2371, respectively) were added and, following 1 h of incubation at room temperature, blots were developed with Western Lightning^®^ Ultra chemiluminescent substrate (PerkinElmer, Inc., Waltham, MA, USA). Changes in the protein level were assessed by densitometric scanning of the bands and corrected for β-actin loading control. The antibody against β-actin (A5316) was obtained from Merck KGaA (Darmstadt, Germany) for Odyssey^®^ CLx imaging system, while the HRP-conjugated anti-β-actin (sc-47778) antibody was obtained from Santa Cruz Biotechnology.

### 4.7. Statistical Analysis

The statistical analysis was performed using GraphPad Prism v 7.05 (GraphPad Software, San Diego, CA, USA) or Microsoft Excel. The data were subjected to Student’s t-test (for two groups), one-way or two-way analysis of variance, and an appropriate post hoc test (ANOVA, Tukey’s test, or Sidak’s multiple comparison test). The data are presented as mean ± S.D. or SEM. Statistically significant differences are denoted with asterisks: * *p* < 0.05, ** *p* < 0.01, *** *p* < 0.001, **** *p* < 0.0001.

## Figures and Tables

**Figure 1 ijms-25-02505-f001:**
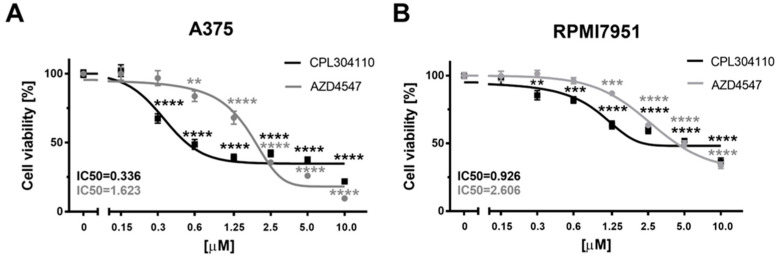
The effect of FGFR inhibitors, CPL304110 and AZD4547, on the viability of human malignant A375 (**A**) and RPMI7951 (**B**) melanoma cell lines. A375 melanoma cells were treated with serial dilutions (0.15–10 µM) of CPL304110 or AZD4547 for 72 h. Data are shown as the mean of three independent experiments (n = 4–6 in each) ± SEM. Statistical significance for a single treatment was estimated using one-way ANOVA and presented as ** *p* < 0.01, *** *p* < 0.001 or **** *p* < 0.0001.

**Figure 2 ijms-25-02505-f002:**
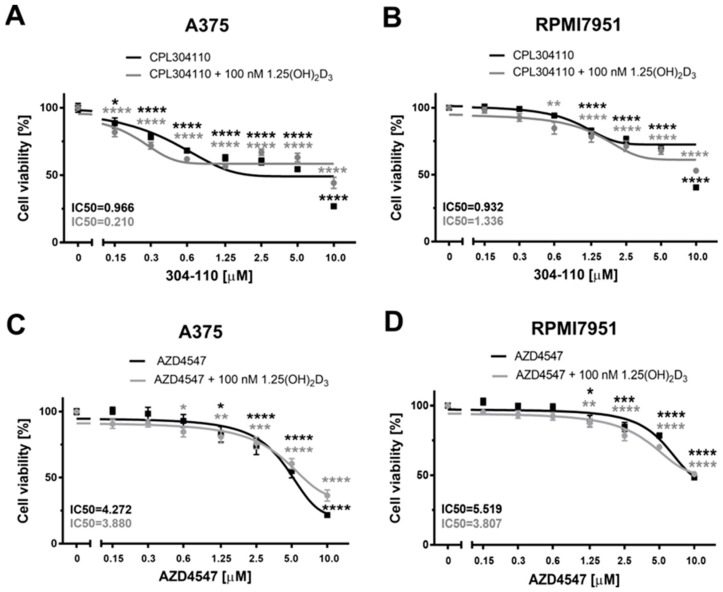
The effect of FGFR inhibitors, CPL304110 (**A**,**B**) and AZD4547 (**C**,**D**), and their combinations with 1,25(OH)_2_D_3_ on viability of the human malignant A375 (**A**,**C**) and RPMI7951 (**B**,**D**) melanoma cell lines. Cells were treated with serial dilutions (0.15–10 µM) of CPL304110 or AZD4547 alone or in combination with 1,25(OH)_2_D_3_ at 100 nM concentration for 48 h. Data are shown as the mean of three independent experiments (n = 4–6 in each) ± SEM. Statistical significance for single treatment was estimated using one-way ANOVA and presented as * *p* < 0.05, ** *p* < 0.01, *** *p* < 0.001 or **** *p* < 0.0001.

**Figure 3 ijms-25-02505-f003:**
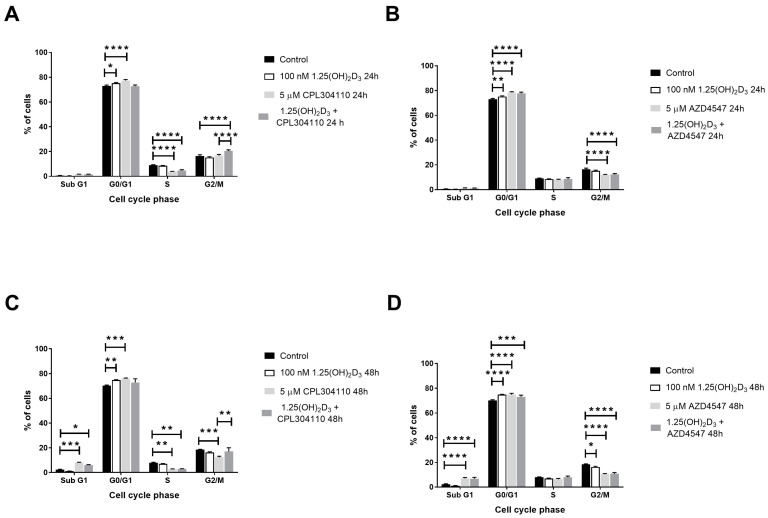
The effect of FGFR inhibitors, CPL304110 (**A**,**C**) and AZD4547 (**B**,**D**) at 5 µM or its combination with 100 nM 1,25(OH)_2_D_3_ on the distribution of human malignant melanoma A375 throughout phases of the cell cycle (SubG1—apoptotic/necrotic cells, G1—growth, S—DNA synthesis, G2/M—preparation for mitosis/mitosis) after 24 or 48 h incubation. Cells were harvested, stained with propidium iodide and analysed by Flow Cytometry. The same control and 1,25(OH)_2_D_3_ data are plotted in an appropriate incubation time graph (for 24 h—(**A**,**B**); and for 48 h—(**C**,**D**)). The data are presented as the mean ± standard deviation (n = 3). Statistical significance was estimated using two-way ANOVA followed by Tukey’s multiple comparison test and presented as * *p* < 0.05; ** *p* < 0.01; *** *p* < 0.001, **** *p* < 0.0001.

**Figure 4 ijms-25-02505-f004:**
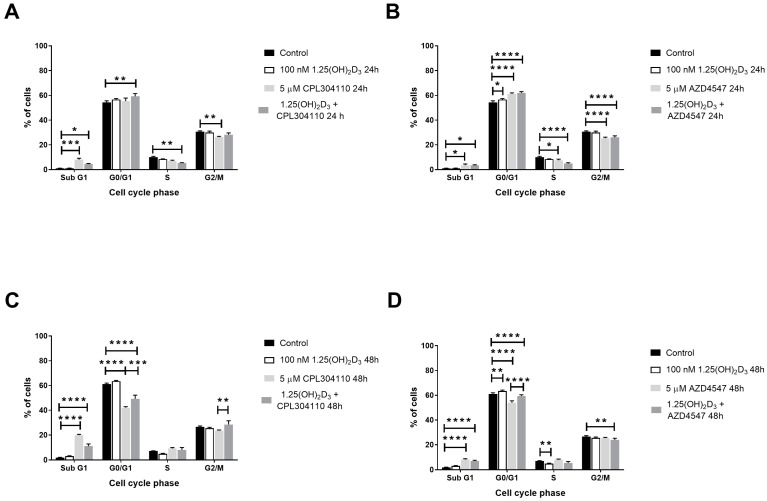
The effect of FGFR inhibitors, CPL304110 (**A**,**C**) and AZD4547 (**B**,**D**) at 5 µM or their combination with 100 nM 1,25(OH)_2_D_3_ on the distribution of human malignant melanoma RPMI7951, throughout phases of the cell cycle (SubG1—apoptotic/necrotic cells, G1—growth, S—DNA synthesis, G2/M—preparation for mitosis/mitosis) after 24 or 48 h incubation. Cells were harvested, stained with propidium iodide and analysed by Flow Cytometry. The same control and 1,25(OH)_2_D_3_ data are plotted on an appropriate incubation time graph (for 24 h—(**A**,**B**); and for 48 h—(**C**,**D**)). The data are presented as the mean ± standard deviation (n = 3). Statistical significance was estimated using two-way ANOVA followed by Tukey’s multiple comparison test and presented as * *p* < 0.05; ** *p* < 0.01; *** *p* < 0.001, **** *p* < 0.0001.

**Figure 5 ijms-25-02505-f005:**
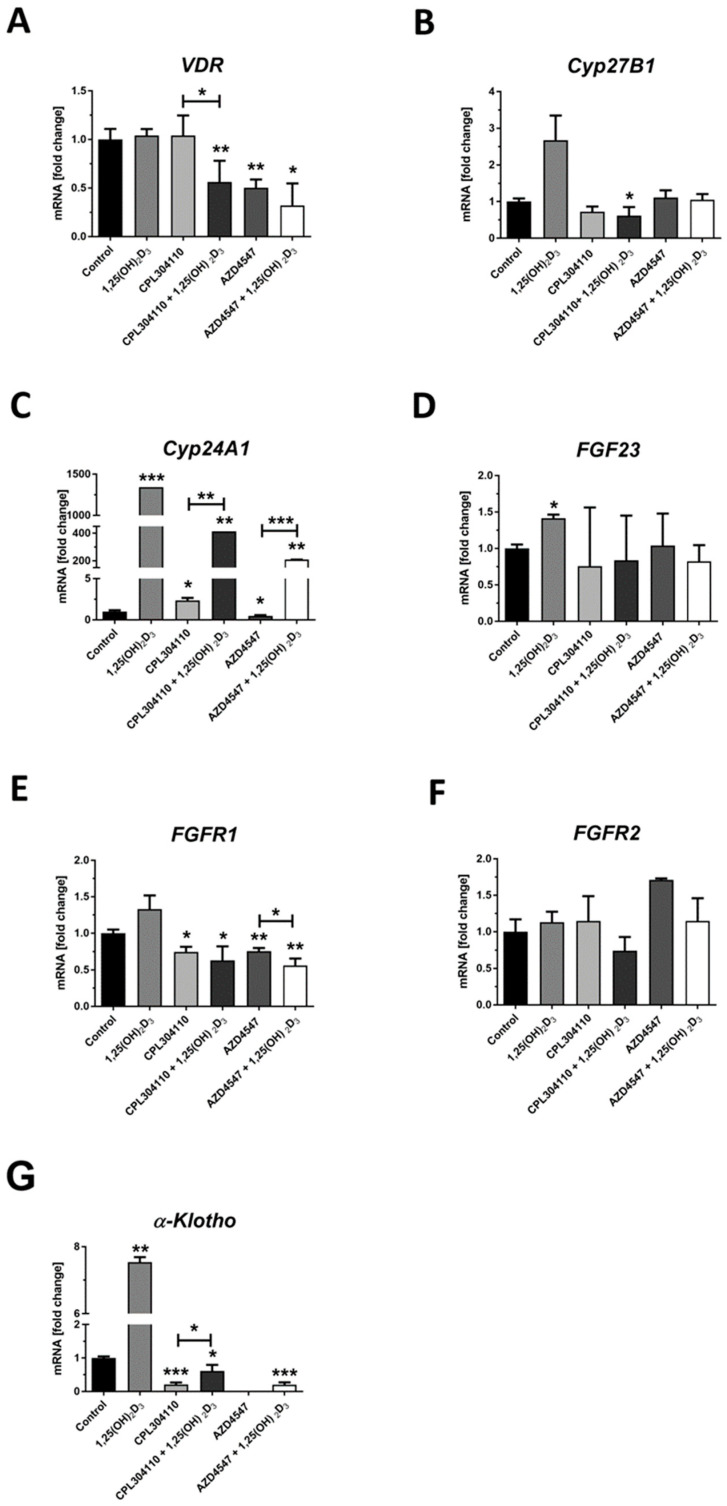
The effect of FGFR inhibitors, CPL304110 and AZD4547, at 5 µM, or their combination with 100 nM 1,25(OH)_2_D_3_ on VDR (**A**), CYP27B1 (**B**), CYP24A1 (**C**), FGF23 (**D**), FGFR1 (**E**), FGFR2 (**F**) and α-KLOTHO (KL) (**G**) gene expression in A375 after 24 h incubation. mRNA levels were measured by qPCR. The results are representative of three experiments carried out in duplicate. * *p* < 0.05, ** *p* < 0.01 and *** *p* < 0.001 vs. untreated control or between the two groups indicated by the bracket.

**Figure 6 ijms-25-02505-f006:**
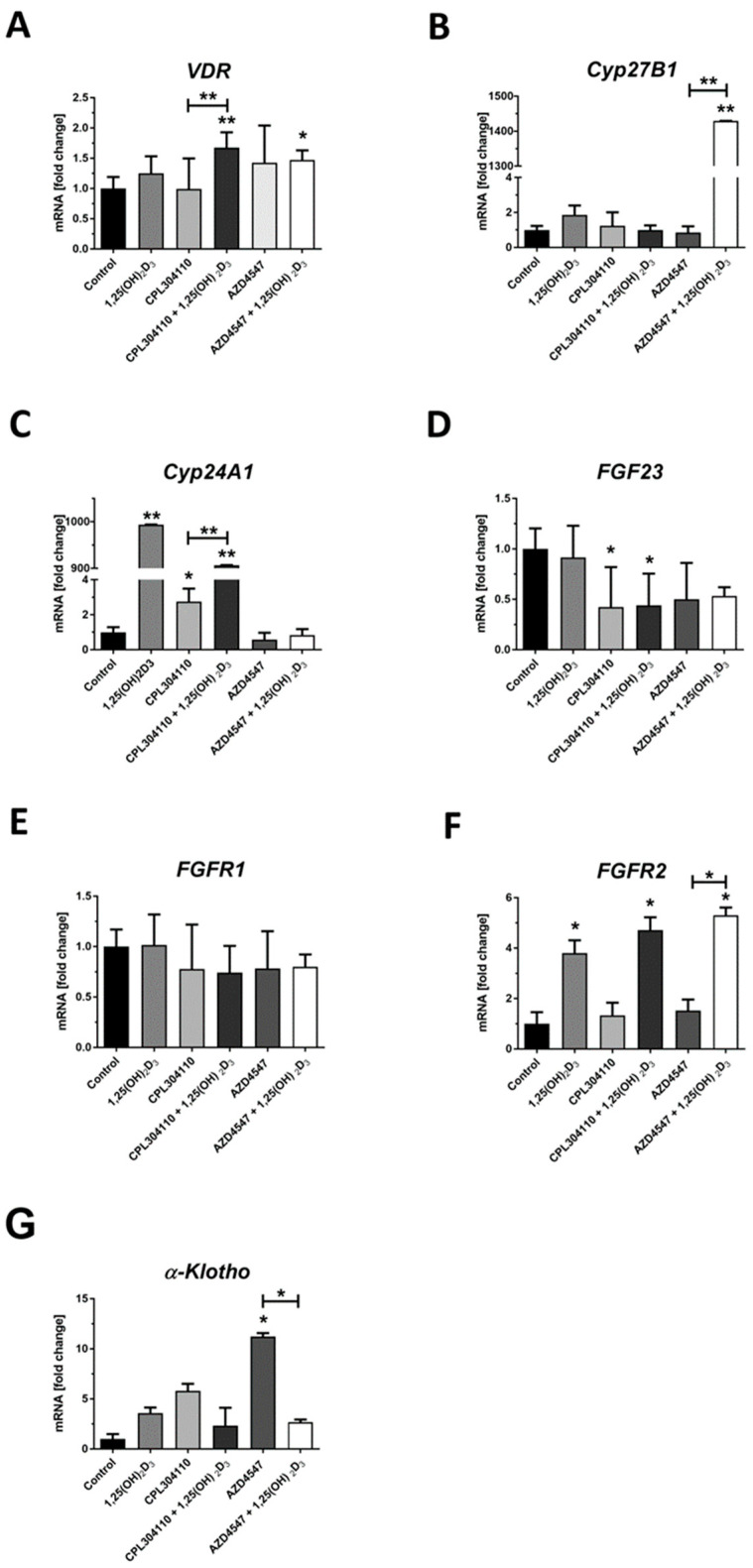
The effect of FGFR inhibitors, CPL304110 and AZD4547, at 5 µM, or their combination with 100 nM 1,25(OH)_2_D_3_ on VDR (**A**), CYP27B1 (**B**), CYP24A1 (**C**), FGF23 (**D**), FGFR1 (**E**), FGFR2 (**F**) and *α-KLOTHO* (KL) (**G**) gene expression in RPMI7951 after 24 h incubation. mRNA levels were measured by qPCR. The results are representative of three experiments carried out in duplicate. * *p* < 0.05 and ** *p* < 0.01 vs. untreated control or between the two groups indicated by the bracket.

**Figure 7 ijms-25-02505-f007:**
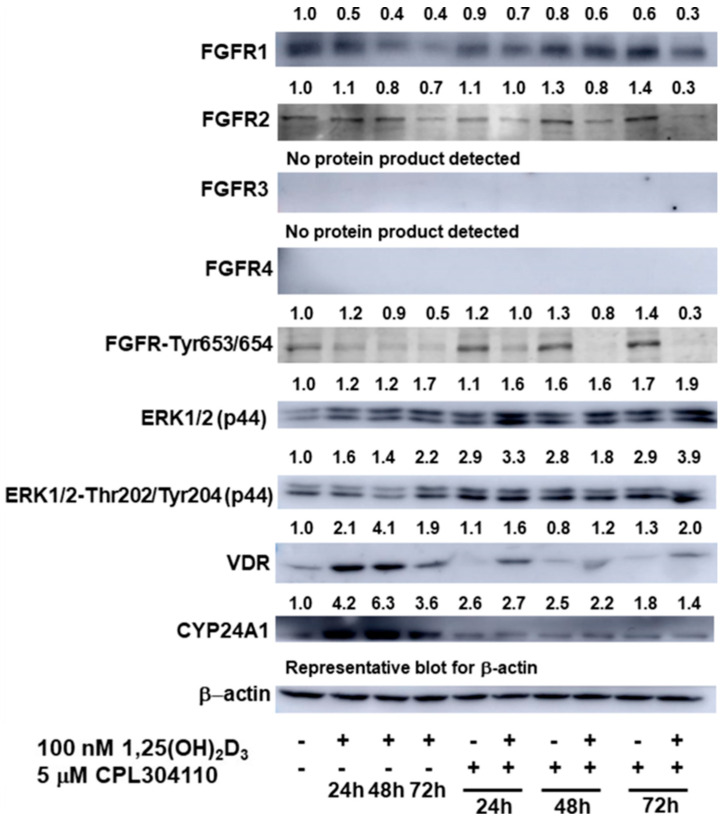
The effect of FGFR inhibitor CPL304110 at 5 µM, or its combination with 1,25(OH)_2_D_3_ at 100 nM concentration on FGFR1-4, phosphorylated at Tyr653/654 FGFR, ERK1/2 (p44), phosphorylated at Thr202/Tyr202 ERK1/2 (p44), VDR and CYP24A1 protein level in A375 during 24 h–72 h incubation. Protein levels were measured by Western blotting, with β-actin used as a control. Data are shown as representative blots from three independent experiments. The visualised bands’ densitometric values presented above are calculated as ratios to appropriate β-actin (not shown in Figure) and are further normalised to nontreated “control” samples, which are referred as “1,0”. β-actin blot presented in the figure is representative.

**Table 1 ijms-25-02505-t001:** Effect of 1,25(OH)_2_D_3_ and CPL304110 or AZD4547 on the distribution of A375 or RPMI7951 melanoma cells through the cell cycle. Data are presented as mean ± S.D. (n = 3).

	SubG1	G1	S	G2/M
A375 24 h				
Control	0.57 ± 0.11	73.01 ± 0.64	9.03 ± 0.33	16.43 ± 0.94
1,25(OH)_2_D_3_	0.37 ± 0.04	75.14 ± 0.60	8.47 ± 0.28	15.18 ± 0.60
CPL304110	1.57 ± 0.15	77.26 ± 0.91	3.73 ± 0.24	16.80 ± 0.81
CPL304110 + 1,25(OH)_2_D_3_	1.56 ± 0.20	72.78 ± 0.93	4.69 ± 0.64	20.41 ± 0.93
AZD4547	1.31 ± 0.18	78.56 ± 0.57	7.93 ± 0.50	11.86 ± 0.46
AZD4547 + 1,25(OH)_2_D_3_	1.09 ± 0.29	77.77 ± 0.96	8.55 ± 1.10	12.38 ± 0.60
A375 48 h				
Control	2.30 ± 0.34	70.23 ± 0.48	7.00 ± 0.30	16.33 ± 0.56
1,25(OH)_2_D_3_	1.03 ± 0.17	74.79 ± 0.33	11.99 ± 1.17	20.13 ± 1.34
CPL304110	8.03 ± 0.35	75.97 ± 0.38	2.61 ± 0.34	12.37 ± 0.89
CPL304110 + 1,25(OH)_2_D_3_	6.06 ± 0.38	72.79 ± 3.02	3.07 ± 0.06	17.10 ± 2.97
AZD4547	7.07 ± 0.66	74.81 ± 1.06	6.39 ± 0.61	10.48 ± 0.53
AZD4547 + 1,25(OH)_2_D_3_	6.87 ± 1.05	73.11 ± 1.03	7.90 ± 1.04	10.84 ± 0.97
RPMI7951 24 h				
Control	1.06 ± 0.13	54.28 ± 1.30	10.09 ± 0.64	30.74 ± 0.59
1,25(OH)_2_D_3_	1.06 ± 0.19	56.60 ± 0.86	8.48 ± 0.19	29.89 ±1.20
CPL304110	8.00 ± 1.20	55.32 ± 2.45	7.03 ± 0.58	26.20 ± 0.71
CPL304110 + 1,25(OH)_2_D_3_	4.41 ± 0.48	59.27 ± 2.15	5.22 ± 0.37	28.23 ± 1.44
AZD4547	3.53 ± 1.01	61.18 ± 0.81	7.46 ± 1.00	25.32 ± 0.89
AZD4547 + 1,25(OH)_2_D_3_	3.41 ± 0.50	61.91 ± 1.07	4.85 ± 0.72	26.07 ± 1.24
RPMI7951 48 h				
Control	1.94 ± 0.18	61.07 ± 0.87	7.09 ± 0.20	26.52 ± 0.78
1,25(OH)_2_D_3_	3.09 ± 0.29	63.41 ± 0.74	4.89 ± 0.40	25.49 ± 0.74
CPL304110	19.81 ± 0.70	42.07 ± 0.81	9.20 ± 0.67	23.52 ± 0.50
CPL304110 + 1,25(OH)_2_D_3_	11.07 ± 1.67	49.30 ± 2.86	8.11 ± 1.79	28.47 ± 2.99
AZD4547	8.17 ± 0.63	53.98 ± 1.53	8.06 ± 0.47	25.37 ± 0.56
AZD4547 + 1,25(OH)_2_D_3_	7.03 ± 0.58	59.35 ± 1.04	5.42 ± 1.15	23.87 ± 1.33

**Table 2 ijms-25-02505-t002:** Primer sequences.

Gene	Forward Primer (5′–3′)	Reverse Primer (5′–3′)
*RPL37A*	CCATTTCAGGCGGCGGTAGTCTT	ACGGTGTCTTTCTCGTTCAC
*VDR*	AGTAAGTGTGCTTGACCTCC	GAGAAGTCTGTACTACCTGAGC
*CYP27B1*	AGACTCGTTTACGTTTGT	GACATACTCGAGAGGGCC
*CYP24A1*	TTTAGACGTGATCCGACG	GTTGTCAAGACCCACTTA
*FGF23*	CGACACCGACACATGTCCAC	TAGTCTCCTACGACCGAAAC
*FGFR1*	CATCAGCTACACCACTTACAGG	TCTCACTACACACCAGAAAGC
*FGFR2*	GAAACAGTTAAGGGTGACGAAG	GGAAAGACTAGACCACAGTCTC
*α-KLOTHO*	CAATAAGAAGTACGAGAGCCCT	GAAACCTGGGTGGAACTCAA

## Data Availability

Data are contained within the article or Appendix A.

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
