# Peer review of "Fibroblast Growth Factor Receptor Inhibitors Decrease Proliferation of Melanoma Cell Lines and Their Activity Is Modulated by Vitamin D"

_ijms, 2024, doi:10.3390/ijms25052505_

Round 1

Reviewer 1 Report

Comments and Suggestions for Authors

A study by Piotrowska et al investigates possible crosstalk between vitamin D-activated pathways and activity of FGFR inhibitors, which should be considered in the further clinical studies. They concluded that CPL110, a novel selective FGFRs inhibitor, is highly efficient against proliferation of primary A375 and metastatic RPMI7951 melanoma cell lines, especially during prolonged incubation, although malignant RPMI7951 melanoma cell line is less sensitive to the tested compounds. 1,25(OH)2D3 slightly, but significantly, enhanced cytotoxic effect of CPL110, decreasing the IC50 value by more than 4-fold for this FGFR inhibitor in A375 cell line.

Specific comments:

1. Figures - why CPL110 is shown in the graph legend as 304-110, while being CPL110 in other parts of the manuscript?

2. "AZD4547 treatment also decreased mRNA level for VDR (p<0.01), however the effect was not further modulated by 1,25(OH)2D3 (Figure 5A)." Is this conclusion affected by relatively high SD and low reproducibility of the biological replicates, isn't it?

3. Primer sequnces are usually given from 5' to 3' end.

4. Fig. 5 and 6 do not need to be shown separately as they present the same results for two different cell lines.

5. Did the Authors perform any cell death-specific test e.g., assessing apoptosis induction? 

6. Western blots for RPMI7951 are missing.

Author Response

Dear Reviewer,

We kindly appreciate all your comments. We believe it has significantly improved our manuscript.

1) Please report genes in Italic style (example: FGFR).

Thank you for the question.  All genes have been reported in Italic style in the final version of manuscript.

2) For the authors, what could be a potential mechanisms of melanoma evasion (in vivo) to FGFR-inhibitor therapy?

Thank you for the question. As the potential mechanisms of secondary resistance (acquired resistance) are still not clear, we can rely on literature. The resistance to FGFR TKIs can develop as a result of gatekeeper mutations within the FGFR protein. It can be also hyperactivation of alternate mitogenic signaling pathways. (Krook, Reeser, Ernst et al., 2021)

This phenomenon was discuss as followed:

“On the other hand, resistance to FGFR inhibitors may develop as a result of receptor mutation or hyperactivation of alternative signaling that overcomes their inhibitory effects.[10]” Page 21

3) Typo error: “Finally, the impact of the experimental treatment of 389 A375 cells on protein level of vitamin D receptor( VDR)”, please correct it in “Finally, the impact of the experimental treatment of 389 A375 cells on protein level of vitamin D receptor (VDR),”

Thank you for the comment. This sentence has been corrected in the manuscript according to your suggestion.

4) I suggest to the authors to report Figure 6 (the bars) in different colors to make it more “legible”.

Thank you for the query. Different colors have been applied for the bars in figure 6 and  figure 5 as well. The figures were changed in the manuscript.

Reviewer 2 Report

Comments and Suggestions for Authors

Thanks to the authors and IJMS staff for allowing me to review this article, for which I recommend acceptance after very minor revision.

1) Please report genes in Italic style (example: FGFR).

2) For the authors, what could be a potential mechanisms of melanoma evasion (in vivo) to FGFR-inhibitor therapy?

3) Typo error: “Finally, the impact of the experimental treatment of 389 A375 cells on protein level of vitamin D receptor( VDR)”, please correct it in “Finally, the impact of the experimental treatment of 389 A375 cells on protein level of vitamin D receptor (VDR),”

4) I suggest to the authors to report Figure 6 (the bars) in different colors to make it more “legible”.

Author Response

Dear Reviewer

Thank you for your comments, which allow us to improve our manuscript.

Specific comments:

  1. Figures - why CPL110 is shown in the graph legend as 304-110, while being CPL110 in other parts of the manuscript?

Thank you for your question. The name of the substance has been unified in all the figures and in the text of the manuscript as well.

  1. "AZD4547 treatment also decreased mRNA level for VDR (p<0.01), however the effect was not further modulated by 1,25(OH)2D3 (Figure 5A)." Is this conclusion affected by relatively high SD and low reproducibility of the biological replicates, isn't it?

Thank you for your valuable comment. Indeed, reproducibility of biological replicates was low, thus standard deviation was high.

  1. Primer sequnces are usually given from 5' to 3' end.

Thank you for the comment. Primer sequences have been corrected given from 5’ to 3’ end. The corrected table was attached to the manuscript.

  1. Fig. 5 and 6 do not need to be shown separately as they present the same results for two different cell lines.

Thank you for the suggestion. As figures 5 and 6 represent expression of several genes for clearer layout we decided to show results for each cell line within a separate graph. The same applied for figures 3 and 4 representing distribution through cell cycle phases. Otherwise, the figure would have panels from A to N, and would be hard to read.

  1. Did the Authors perform any cell death-specific test e.g., assessing apoptosis induction? 

Thank you for the remark. We didn’t perform any death-specific test. However, within cell cycle analysis, SubG1 fraction corresponds to apoptotic/necrotic cells.

Our observation on the given subject were included in the abstract”

“Both tested FGFR inhibitors triggered G0/G1 cell cycle arrest in A375 melanoma cells and increased apoptotic/necrotic SubG1 fraction in RPMI7951 melanoma cells. “

In the result section Point 2.2

To highlight the relationship between the SubG1 fraction and apoptosis/necrosis, a relevant fragment has been added to this section:

“Interestingly, simultaneous treatment of A375 melanoma cells with CPL304110 and calcitriol triggered rather G2/M cell cycle arrest (20,41% vs 16,43% in control cells, p<0.0001, Fig. 3A) indicating induction of apoptotic/necrotic cell death.”

And the induction of apoptosis has been discussed already in the manuscript:

“Additionally, our observations of melanoma cells death induction after treatment with CPL304110 or AZD4547 are in an agreement with other published data, that described induction of apoptosis after inhibition of FGFR system [36,37]. Interestingly, 1,25(OH)2D3 modulated the activity of CPL304110, but not AZD4547, as to the regulation of the cell cycle in melanoma cells since we observed G2/M rather than G0/G1 cell cycle arrest in A375 cells and G0/G1 cell cycle arrest rather than an induction of SubG1 fraction in RPMI7951 melanoma cells compared to the monotreatment.” Page 22.

  1. Western blots for RPMI7951 are missing.

Thank you for your remark. As a A375 melanoma cell line was more susceptible to CPL304110 inhibitory effect than RPMI7951 in proliferation tests and cell cycle analysis, we decided to focus on FGFR signaling solely in that cell line.

[1]           Krook, M.A., Reeser, J.W., Ernst, G., Barker, H., Wilberding, M., Li, G., Chen, H.Z. and Roychowdhury, S., 2021. Fibroblast growth factor receptors in cancer: genetic alterations, diagnostics, therapeutic targets and mechanisms of resistance, Br J Cancer. 124, 880-892.

Round 2

Reviewer 1 Report

Comments and Suggestions for Authors

The manuscript has been improved.